# Methane emissions from US low production oil and natural gas well sites

Mark Omara [1✉], Daniel Zavala-Araiza [1,2], David R. Lyon [1], Benjamin Hmiel[1], Katherine A. Roberts[1] & Steven P. Hamburg[1]

Eighty percent of US oil and natural gas (O&G) production sites are low production well sites, with average site-level production ≤15 barrels of oil equivalent per day and producing only 6% of the nation's O&G output in 2019. Here, we integrate national site-level O&G production data and previously reported site-level $CH_4$ measurement data ($n = 240$) and find that low production well sites are a disproportionately large source of US O&G well site $CH_4$ emissions, emitting more than 4 (95% confidence interval: 3—6) teragrams, 50% more than the total $CH_4$ emissions from the Permian Basin, one of the world's largest O&G producing regions. We estimate low production well sites represent roughly half (37—75%) of all O&G well site $CH_4$ emissions, and a production-normalized $CH_4$ loss rate of more than 10%—a factor of 6—12 times higher than the mean $CH_4$ loss rate of 1.5% for all O&G well sites in the US. Our work suggests that achieving significant reductions in O&G $CH_4$ emissions will require mitigation of emissions from low production well sites.

[1] Environmental Defense Fund, Austin, TX 78701, USA. [2] Institute for Marine and Atmospheric Research Utrecht, Utrecht University, 3584 CC Utrecht, The Netherlands. ✉email: momara@edf.org

Mitigation of methane ($CH_4$) emissions, a powerful greenhouse gas with >80× the 20-year warming potential of carbon dioxide[1,2], is widely recognized as strategically integral to the attainment of the climate-neutrality goals of Paris Agreement[3,4]. In the United States, official estimates from the US Environmental Protection Agency (EPA) indicate nearly one-third (30%) of anthropogenic $CH_4$ emissions arise from oil and natural gas (O&G) operations[5]. However, a large body of measurement-based studies[6–15] have consistently found higher O&G $CH_4$ emissions than is estimated in EPA inventories. Alvarez et al.[16] synthesized research on US O&G $CH_4$ emissions in 2015 and found 13 teragrams (1 Tg = 1 million metric tons), 60% higher than the Greenhouse Gas Inventory (GHGI) estimates for 2015 as estimated in 2017; in Reporting Year 2021, EPA lowered estimated 2015 emissions making the difference 70%[5]. Much of this discrepancy has been attributed to the O&G production sector, where measurement-based estimates are ~2× higher than the GHGI[16–18], with recent research suggesting substantial underestimation in the GHGI attributed to fugitive emissions from well site equipment and unintentional emissions from liquids storage tanks[18].

The US O&G production sector is diverse and complex, with over 800,000 active onshore O&G production wells in 2019[19]. Methane emissions at O&G production well sites—which may have one or multiple wellheads—arise from sources that are common throughout O&G operations (e.g., fugitive emissions from leaking valves and connections and vented emissions from storage tanks and pneumatic devices), in addition to nonroutine sources characterized by excessive, unintentional emissions. Measurement-based studies have generally found weak correlations of $CH_4$ emissions with site-specific parameters, including O&G production rates, water production, or site age[12,20–22]. However, O&G production declines substantially over the first few years in the life of the well, such that the number of new, high-productivity wells represents a small percentage of the total number of operating wells, where older, low-productivity wells dominate. As a result, production characteristics of US O&G wells are highly skewed: >90% of the nation's O&G production comes from ~20% of wells[19].

Furthermore, a key characteristic of measurement-based O&G site-level $CH_4$ emissions is the heavy-tailed distributions[8,9,12,13,17,23], where a small fraction of sites is responsible for a disproportionately large fraction of total $CH_4$ emissions. While the skewness in the distributions of O&G site-level $CH_4$ emissions and production characteristics are well known, their effect on the national distribution of aggregate $CH_4$ emissions among low- and high-productivity O&G production sites has received little scrutiny and is much more uncertain.

We define a well site's total O&G production in units of barrels of oil equivalent per day (boed), a single metric representing the site's combined oil (barrels produced) and gas (1 boe = 6 thousand cubic feet, Mcf)[24] production averaged over the well site's total production days in the year. We focus on the low production well site category, where each site has a combined O&G production rate averaged over the year of ≤15 boed[25]. We then use available O&G production data from proprietary sources[19] to assess the regional distribution, O&G production characteristics, and operator profiles for low production sites. Using these data in combination with data on low production well site $CH_4$ emissions previously collected from a diversity of regions across the United States, we generate a new national estimate of their total $CH_4$ emissions and assess the significance of these emissions relative to $CH_4$ emissions from all US O&G production sites. This assessment carries significant policy implications for the effective mitigation of US O&G $CH_4$ emissions.

## Results and discussion

**Characteristics of US low production oil and gas well sites.** We use the O&G well- and production data from Enverus Prism[19], a commercial platform which collects and aggregates public and proprietary O&G data, to assess the production, age, and operator profiles of low production well sites. We consider each low production site with reported production data as a commercially viable production site or site that routinely produces O&G products that are used for energy consumption. A low production well site may have one or multiple wellheads (average 1.03 wells per site; Methods) with O&G processing equipment that may include separators, dehydrators, pneumatic devices, compressors, flare stacks, and/or hydrocarbon liquids storage vessels[10,18,22]. In 2019, we estimate that 565,000 (3 sf, Methods) low production well sites accounted for 81% of the total number of US active onshore O&G well sites. Yet, they accounted for a substantially smaller share of national oil (5.9%), gas (5.5%), and combined O&G (5.6%) production (Fig. 1).

We classify national low production sites into four cohorts of site-level production rates: (i) >0–2, (ii) 2–5.4, (iii) 5.4–9.7, and (iv) 9.7–15 boed (see Methods, Supplementary Note 6 for further discussion). A majority of low production well sites (57%), 46% of active onshore US O&G well sites, produce very little O&G, ≤2 boed/site, with cumulative production of just 0.7% of total US O&G production, representing 12% of total O&G production from all low production sites. We refer to this subset of low production sites as ultralow production sites and discuss their significance in the following sections.

There is regional diversity in the production characteristics of low production well sites (Fig. 1), with the predominantly gas-producing Appalachian region (Region 1 in Fig. 1) being notable for its large abundance (i.e., 90%, $n = 160,000$) of ultralow production well sites (Fig. 1d). Among all low production well sites, these Appalachian ultralow production sites represent 29% of US low production well sites and 5.4% of total O&G production from low production sites (Fig. 1c).

The distribution for site age, defined as the mean number of years in production as of December 2019, shows little variability across regions (Fig. 1e). The mean age for the ultralow production sites is 25 years, only slightly higher than that for sites producing >2 boed/site at 21 years. In general, about 10% of all low production well sites ($n = 73,000$) are ≤10 years old (Fig. 1e) with combined O&G production representing 20% of total production from low production well sites, indicative of average declining production with age.

Oil and gas production at newly drilled and completed wells exhibits a rapid rate of decline following initial production. We assessed the production history of over 44,000 single-well low production well sites that were actively producing in 2019 and had their first reported production date in the years between 2012 and 2019. We find that, on average, the initial site-level production for single-well O&G production sites that are vertically drilled is ~20 boed/site, ramping up to ~25 boed/site within the first three months of production, before exponentially declining to below the low production well site productivity threshold of 15 boed within generally 1 to 2 years. For horizontally-drilled wells, we estimate an average initial production of 100 boed/site, with a ramp-up to ~150 boed/site within the first three months and declining to below 15 boed within ~3 to 5 years (Supplementary Note 3). This average boed decline profile for single-well sites suggests continued and rapid growth in the number of future low production well sites, tempered by the rate of growth in the number of newly completed O&G wells and the rate at which operators plug and/or abandon these wells.

There are more than 11,000 O&G operators nationally (Fig. 2a). While a significant proportion (6100 operators, or 52%) own

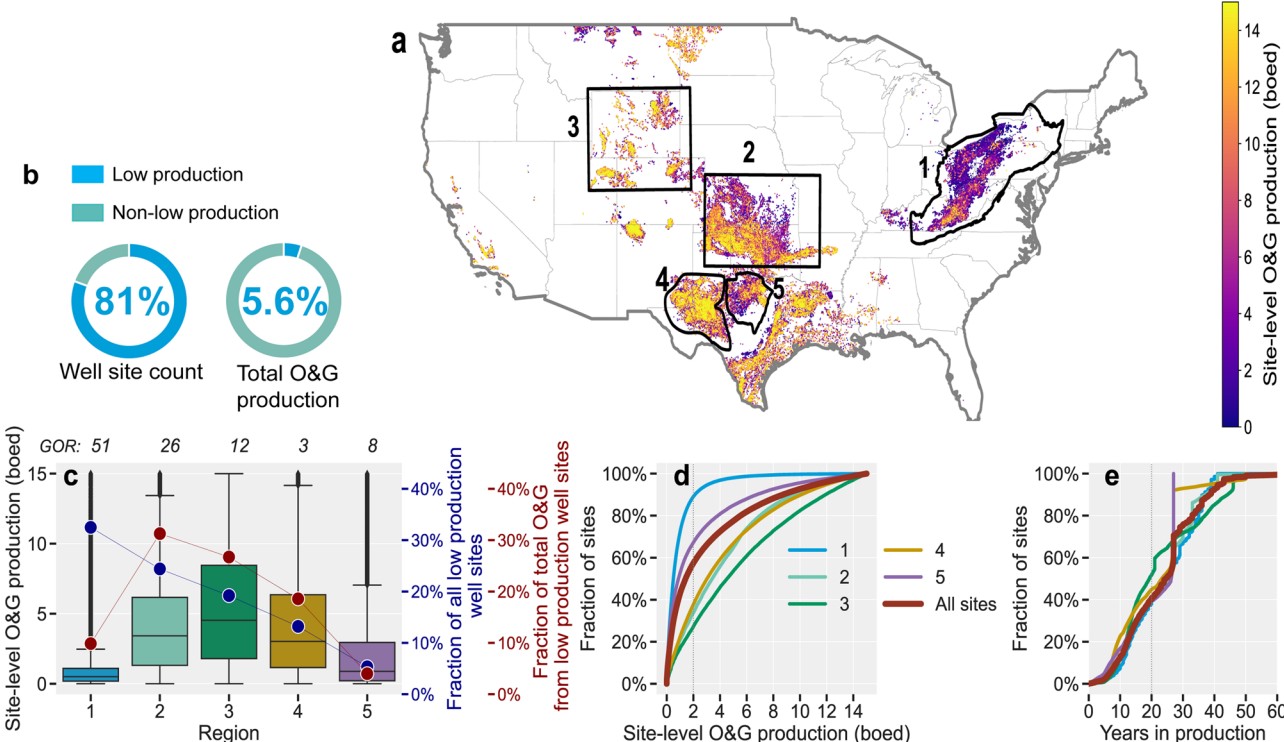

**Fig. 1 Characteristics of US low production oil and gas well sites. a** Spatial distribution of active onshore low production well sites ($n = 565,000$) color-coded by site-level O&G production in barrels of oil equivalent per day (boed) per site. The numbered boxes show a few of the major low production well site regions, including those for which site-level $CH_4$ emissions data are available: (1)—Appalachian, (2)—Oklahoma/Kansas/Arkansas, (3)—Colorado/Utah/Wyoming, (4)—Permian Basin, and (5) Barnett Shale. **b** Distribution of the national number of well sites and O&G production, comparing low production sites with non-low production sites. **c** Box plots (centerline, median; box limits, upper and lower quartiles; whiskers, 1.5× interquartile range; points, outliers) showing the distribution of site-level O&G production in each of the five O&G production regions with large numbers of low production well sites shown on the map. The average gas-to-oil ratio (GOR, Mcf/barrel) is shown on the top x-axis. These five regions account for three-quarters (76%) and two-thirds (68%) of the total number and O&G production from all low production well sites, respectively. The horizontal lines within each box plot show the median production rate per site. On the right y-axis, the percentage of the total count of low production well sites and total O&G production from all low production well sites are shown in blue and red, respectively. **d** Cumulative distribution functions of site-level O&G production for all low production well sites (red line) and well sites in each of the regions shown on the map (blue line—Region 1, light green—Region 2, dark green—Region 3, orange—Region 4, purple—Region 5). **e** Cumulative distribution functions of low production well site age, representing the years in production as of December 31, 2019 and based on the reported first production date. Lines are color-coded as in **d**. Analysis based on data from Enverus Prism[19] for 2019.

≤5 low production well sites each, the majority of low production well sites (77%) and O&G production (83%) are owned by 770 mid-size to large operators with >100 low production well sites each (Fig. 2b, c). For the ultralow production cohort, these same 770 operators also dominate site count (77%) and O&G production (82%) nationally (Fig. 2b). However, there is regional variability in the ownership profile of the ultralow production sites. For example, while the Appalachian sites (Region 1, Fig. 1) are dominated by operators with >100 well sites each, the Barnett sites (Region 5) are dominated by operators with 11–50 well sites each (Fig. 2b).

Among operators that own 1–50 low production well sites, there are consistent patterns in well site characteristics with the ultralow production sites dominating, but the distribution has a long tail that extends to 15 boed/site (Fig. 2f). This result indicates that small operators own low production well sites with a range of site-level production rates (i.e., not only the ultralow production cohort) and underscores that they do not dominate either the low production well site count or total O&G production from low production well sites.

**Methane emissions at low production oil and gas well sites: insights from previous site-level studies.** Previous studies

indicate $CH_4$ emissions at low production well sites arise from sources that are common throughout all O&G production operations, including intentionally vented emissions and unintentional emissions from well site equipment such as wellheads, pneumatic devices, separators, dehydrators, compressors, flare stacks, and/or storage vessels[10,18,22]. (Supplementary Fig. 21). At low production well sites, field observations report a common theme revolving around the issue of well site equipment negligence and disrepair[10,22] as the primary driver of $CH_4$ emissions. Most proximately, recent work by Deighton et al.[22] documents several of these maintenance-related issues, including, for example, (i) leaks at fittings and joints, (ii) leaks and vents from rusted pump jacks, tanks, and other onsite gathering infrastructure, and (iii) evidence of well site neglect or poor maintenance, such as wellheads or casings covered in weeds or fallen trees. In several instances, emissions at low production well sites were reported as "audible", "visible" or with an "oily smell", characteristic of emissions sources likely to be effectively resolved via standard leak detection and repair (LDAR) practices, including Audio, Visual, and Olfactory (AVO) inspections.

In this study, we compile and analyze previously published data on site-level $CH_4$ emissions at low production sites to assess the magnitude and significance of their $CH_4$ emissions relative to total US O&G production site $CH_4$ emissions[16]. We focus on

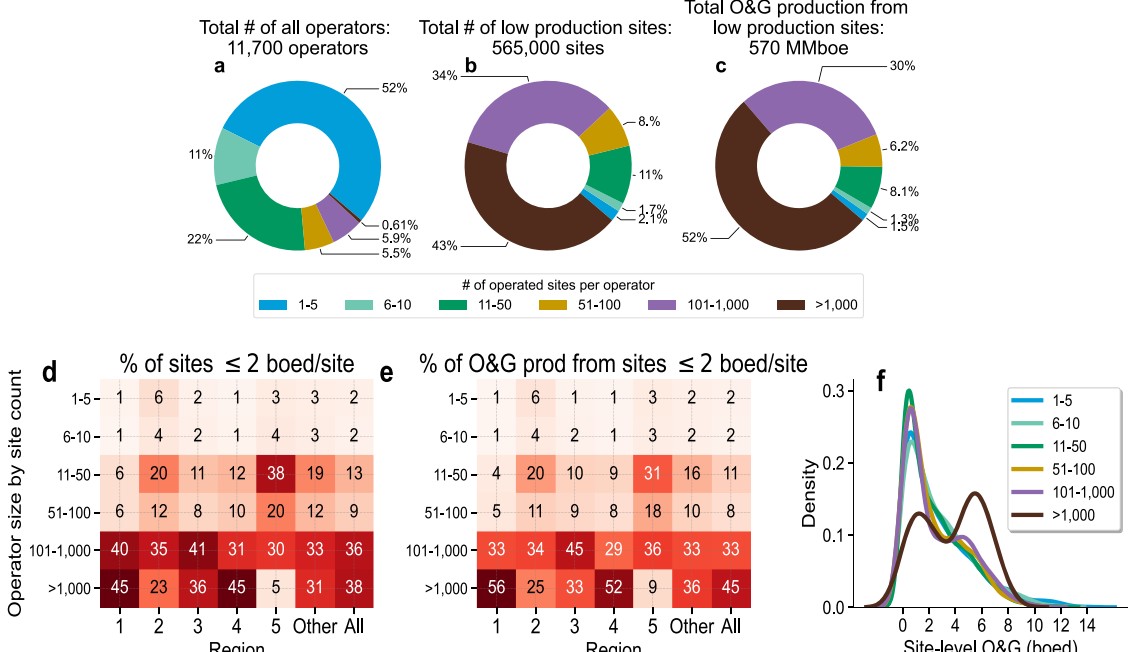

**Fig. 2 Low production well site operator profile. a** Distribution of the total number of all O&G well site operators. **b** Distribution of the number of operated low production well sites by operator size. **c** Distribution of O&G production for operators with 1–5 (blue), 6–10 (light green), 11–50 (dark green), 51–100 (orange), 101–1,000 (purple), and >1,000 operated sites (dark red). **d** Heatmap showing the distribution of well sites and **e** the distribution of O&G production for only the ultralow production sites (producing ≤2 boed/site) and for each operator category within each region shown in Fig. 1. "Other" means all locations not included in Regions 1–5 in Fig. 1 and "All" indicates national statistics for all ultralow production sites. For example, in Region 1, 1% of ultralow production well sites are owned by operators with 1–5 sites each and, for these sites and operators, their combined production accounts for only 1% of the total. **f** Density plot showing similarities in the distribution of mean site-level O&G production for each operator category. For operators with more than 50 operated well sites, a bimodal distribution or the second cluster of sites producing >2 boed/site emerges. Operator names and data are based on Enverus Prism's[19] aggregation into single operator names, including rolling up subsidiaries to the parent company whenever such information is publicly disclosed.

site-level measurement studies, performed using ground-based downwind measurement approaches[10,12,13,17,20,21] that do not require operator-provided access to measured sites and can resolve total $CH_4$ emissions at each measured site, but generally do not resolve source-specific emissions (Methods). Our sample of 240 site-level $CH_4$ emissions data for low production sites is drawn from six independent studies[10,12,13,17,19,20] across six US O&G basins. The most-reported data attributes in these studies are the mean site-level $CH_4$ emission rates (mass of $CH_4$ emitted per hour) and site-level O&G production rates. While limited in size relative to the total population of low production sites, these data are drawn from a diversity of O&G production basins and have broadly representative site-level production rates (range: 0.01–15 boed) and $CH_4$ distribution that support statistically robust estimation of national-scale $CH_4$ emissions (Methods).

We assess $CH_4$ emissions at low production sites on the basis of absolute $CH_4$ emission rates (i.e., the mass of $CH_4$ emitted per hour) and the production-normalized $CH_4$ loss rates (i.e., $CH_4$ emitted relative to $CH_4$ production)—a useful metric for comparing the degree of $CH_4$ loss among different production regions or categories of production sites and can reveal the existence of excessive emissions that may result from avoidable abnormal operating conditions[26].

Our synthesis of the 240 site-level $CH_4$ emission measurements shows a wide range of results, reflecting, in part, the stochastic character of $CH_4$ emissions at these sites. Most low production well sites (75%) have detectable site-level $CH_4$ emissions of up to 5 kg $CH_4$/h (Fig. 3). The unadjusted arithmetic mean $CH_4$ emission rate is 2.6 kg $CH_4$/h/site (95% bootstrap confidence interval on the mean: 1.6–4 kg $CH_4$/h/site) for a weighted-average

$CH_4$ loss rate of 12% of total $CH_4$ production, assuming an average 80% $CH_4$ content in produced natural gas[5]. We note that some of the measured sites in the consolidated dataset ($n = 9$) are oil-only sites, with no reported gas production, but with measured $CH_4$ emissions that range from below the method detection limit (i.e., <0.01 kg $CH_4$/h/site for tracer flux quantification and <0.036 kg $CH_4$/h for OTM-33A quantification; see Methods) to 9 kg $CH_4$/h. The full range of detectable site-level $CH_4$ emissions at low production well sites are within that for all O&G production sites[16,17] but are more than an order of magnitude higher than measured $CH_4$ emissions at unplugged abandoned wellheads[27,28].

The empirical distribution of absolute $CH_4$ emission rates indicates that the top 5% of high-emitting sites are responsible for ~50% of cumulative emissions (Methods), with each site emitting >7.3 kg $CH_4$/h. The data suggest an increased likelihood of high $CH_4$ emission potential for low production well sites producing >~2 boed/site (Fig. 3b). Skewed $CH_4$ emissions distributions have been observed consistently across the O&G supply chain[8,9,12,13,17,23,29]. Although they have stochastic and low-probability occurrence at any one site[8], the significant influence of high-emitting sites is well-documented and is postulated as the primary driver for the observed discrepancy between inventory/bottom-up component-level methods and site-level measurement-based estimates[16].

For low production well sites, we also observe a second dimension to the skewness in the $CH_4$ emissions distribution: among sites with reported gas production, the top 15% of sites based on $CH_4$ loss rates, emit >32% of their $CH_4$ production, while the top 5% exhibit $CH_4$ loss rates of >90%. Furthermore,

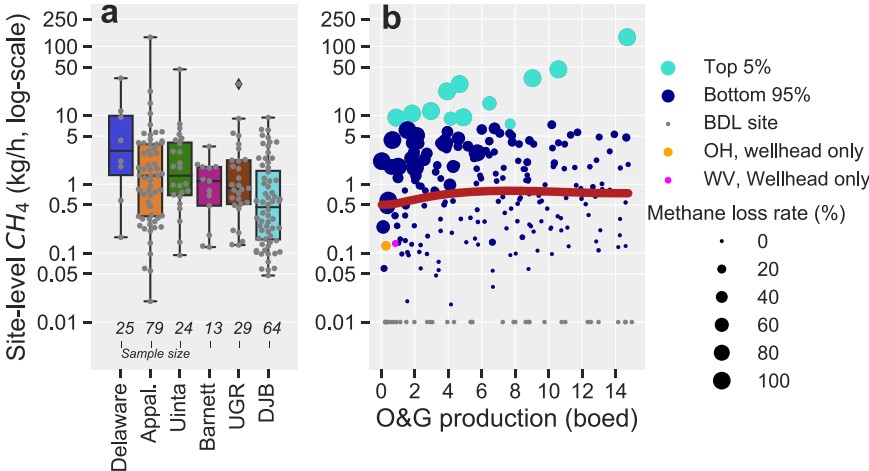

**Fig. 3 Low production well site CH₄ emissions data as reported in previous studies. a** CH₄ emissions data for six basins with at least $n > 5$ observations shown as box plots (centerline, median; box limits, upper and lower quartiles; whiskers, 1.5× interquartile range) and individual points (gray circles). Sample sizes are shown at the bottom of the plot. Only site-level measurements above method detection limits of 0.01–0.036 kg/h are shown. Appal.— Appalachian (Pennsylvania, Ohio, West Virginia); Delaware (Texas/New Mexico); Barnett (Texas); Uinta (Utah); UGR—Upper Green River (Wyoming); DJB—Denver-Julesburg Basin (Colorado). Low production well site data were a subset of site-level measurements reported by: Robertson et al[13]., Robertson et al[21]., Caulton et al[12]., Omara et al[10]., Omara et al[17]. and Brantley et al[20]. **b** Relationship between measured site-level CH₄ emissions and O&G production in barrels of oil equivalent per day (boed). The plot shows the top 5% of high-emitting sites ($n = 12$, green symbols), the bottom 95% of sites ($n = 192$, blue symbols), and below-detection-limit (BDL) sites ($n = 36$, gray symbols). Each site's CH₄ loss rate is indicated by the size of the circles. Oil-only sites or sites with reported CH₄ loss rates >100% are assigned values of 100%. The orange and pink symbols represent the mean wellhead-only CH₄ and O&G production for low production sites sampled in Ohio[22] and West Virginia[31]. Data from these two studies were not used in emission models because they exclude other sources such as tanks and separators, but are shown here to illustrate that wellhead-only CH₄ emissions can be significant even at low production well sites. The solid-dark red line shows the nonparametric Bayesian regression model for the bottom 95% of sites (see Methods).

there is a tendency toward higher CH₄ loss rates as site-level O&G production declines (Fig. 3b), consistent with previous observations for well sites[16,17] and natural gas production regions[30]. Indeed, two recent studies focused on CH₄ emission characterization at the wellhead exclusive of other site-specific sources (e.g., storage tanks and separators) reported mean CH₄ loss rates of 8.8% (for sites producing ~0–3 boed)[31] and 21% (for sites producing <1 boed)[22] of CH₄ production in West Virginia and Ohio, respectively, showing the significant CH₄ emissions that can occur even from a single source, i.e., wellheads, at low production well sites (Supplementary Fig. 24).

By modeling the temporal evolution of site-level emissions, Cardoso-Saldana and Allen[32] attributes these increasing proportional losses to the interplay between emission sources that are production-dependent and decline rapidly with declines in production (e.g., condensate flashing) and those that are production-independent (e.g., fugitive leaks and venting from pneumatic devices). As site-level production declines over time, there is a substantial increase in the relative contribution of production-independent emission sources, resulting in higher CH₄ loss rates. Assuming the empirical distribution of CH₄ loss rates characterized among the 240 measured sites is representative of national patterns, the data suggest a small fraction of low production well sites (5% or $n = 28,000$) are not just high-emitting (on a mass basis), but "functionally super-emitting"[26] with extremely high CH₄ loss rates indicative of the existence of avoidable abnormal process operating conditions (e.g., malfunctioning processing equipment).

Further evidence for extremely high, but low-frequency CH₄ emissions at low production well sites can be found in recent work by Cusworth et al.[33] which used an aerial screening approach to identify and characterize the persistence of large (>10–20 kg/h) CH₄ sources in the Permian Basin. We spatially linked, and visually confirmed in satellite imagery, the location of

their detected CH₄ plumes to 62 unique low production well site sources within the Permian Basin (Supplementary Note 7). Measured CH₄ emissions at these predominantly oil-production sites ranged from ~50–800 kg CH₄/h, with their cumulative CH₄ emissions far exceeding their reported total CH₄ production by a factor of 30× (see discussion in Supplementary Note 7). While we estimate a very low prevalence rate (~0.05% in the Permian Basin; Supplementary Note 7) for such abnormally high CH₄ emissions among the Permian low production well sites, their existence nevertheless underscores the significant CH₄ waste potential as well as the CH₄ mitigation opportunities at low production well sites.

The stochasticity in the site-level CH₄ emission characteristics[8,22] likely explains, in part, the observed variability in the empirical distribution of basin-level CH₄ emissions (Fig. 3a). Other factors such as operator-specific practices, including voluntary or mandated O&G emission reduction programs, could contribute to observed variability, although these are difficult to quantify with available data. Overall, from the ensemble of basin-level data with $n > 25$ observations, we find statistical similarities in the empirical distribution of site-level absolute CH₄ emissions among measured low production well sites in the Appalachian, Upper Green River, and Denver-Julesburg basins (Methods). This statistical similarity supports our consolidation of data from a diverse set of O&G basins to assess the total CH₄ emissions attributable to the national population of low production well sites.

**National estimate of low production well site methane emissions.** Our assessment of national-level CH₄ emissions from low production sites leverages the broadly representative distribution of site-level production and statistical similarities in basin-scale empirical CH₄ distributions (see Methods) in our consolidated

sample of measurement-based site-level data ($n = 240$). We use these data in a hybrid nonparametric Bayesian regression and Monte Carlo model to separately assess the emissions contribution of the top 5% of sites based on absolute $CH_4$ emissions (green symbols in Fig. 3b), the bottom 95% of sites (blue symbols in Fig. 3b) and the influence of below-detection-limit sites (gray symbols in Fig. 3b, Methods). For the high-emitting sites, we develop frequency and emissions distributions based on random nonparametric bootstrap resampling. For the bottom 95% of sites with detectable emissions, we develop site-level emission factors based on a nonparametric Bayesian regression model (solid-dark red line in Fig. 3b) of the site-level $CH_4$ emissions as functions of site-level O&G production. This approach accounts for the empirically observed relative independence of site-level $CH_4$ emissions with O&G production for sites producing ~>2–4 boed/ site and an apparent declining trend in absolute site-level $CH_4$ emissions for the ultralow production sites (Supplementary Fig. 15). Finally, we develop a frequency distribution for the below-detection-limit sites and use this distribution to decrement the modeled site-level $CH_4$ emissions for the bottom 95% of sites (Methods).

Our estimate for total $CH_4$ emissions from active onshore low production O&G well sites in 2019 is 4 Tg (1 s.f.), with a 95% confidence interval (CI) on the mean of 3–6 Tg (Fig. 4a). The mean estimate is 54% (95% CI: 37–75%) of the 7.6 Tg for total O&G $CH_4$ emissions from all O&G production sites based on Alvarez et al.[16], which we consider the best current measurement-based estimate of national-scale $CH_4$ emissions from all US O&G production sites. Our measurement-based estimate for all US low production well sites is roughly 50% more than the total $CH_4$ emissions from the entire Permian Basin (2.7 Tg)[14], one of the world's largest O&G producing regions. Additionally, the 4 Tg of low production well site $CH_4$ emissions is >10% greater than the US EPA's estimate of ~3.4 Tg for all US O&G production site $CH_4$ emissions in 2019[5]. These $CH_4$ emissions are equivalent to $CH_4$ loss rates of 13% (95% CI: 8–17%) relative to $CH_4$ production in 2019, assuming 80% $CH_4$ content in produced natural gas. This $CH_4$ loss rate is a factor of 6–12 times higher than the mean $CH_4$ loss rate of 1.5% for all O&G well sites based on Alvarez et al.[16] (Fig. 4a).

We estimate that ~50% (95% CI: 20–80%) of low production well site $CH_4$ emissions are from the top 5% of sites that emit

>7 kg $CH_4$/h/site, consistent with the empirical distribution and with previous results from a large body of O&G $CH_4$ studies[8,9,12,17,23,26,29,33]. Overall, our modeling indicates that 90% of low production well sites emit an average of <1 kg $CH_4$/h/site, while 50% emit >10% of their $CH_4$ production (Fig. 4c). Based on a total of 4 Tg $CH_4$ emitted by 565,000 low production well sites in 2019, we estimate an average site-level $CH_4$ emission rate of 0.8 kg/h/site (95% CI: 0.5–1.2). This site-level estimate for low production well sites is approximately 50% lower than the mean site-level $CH_4$ emission rates for all US natural gas production sites (1.7 kg $CH_4$/h/site[17]). Thus, while mean low production well site emissions are lower than that for all O&G production sites on an absolute basis, their production-normalized $CH_4$ loss rates are significantly higher, consistent with previous assessments focused on $CH_4$ emissions from US natural gas production sites[17].

We find that the ultralow production cohort accounts for 25% (95% CI: 17–49%) of total low production site $CH_4$ emissions (Fig. 4a), representing ~10% of total US O&G $CH_4$ emissions from production sites and only 0.7% of US O&G production. In addition, the Appalachian region dominates regional $CH_4$ emissions, with an estimated total of 1.2 Tg (95% CI: 0.8–1.9; Fig. 4b). We estimate the ultralow production sites (i.e., sites ≤2 boed) in the Appalachian account for ~one-half (95% CI: 40–60%) of the region's total low production well site $CH_4$ emissions, where the estimated regional $CH_4$ loss rate is 26% (95% CI: 17–40%; Fig. 4b). These results underscore the significance of the ultralow production sites as sources of O&G $CH_4$ emissions, especially in the Appalachian region where they account for ~90% of all low production sites.

**Policy implications.** Eighty percent of all US O&G production sites are low production sites, yet they produce only 6% of the nation's O&G output. Even as their production declines over time, $CH_4$ emissions at low production well sites continue from both routine and nonroutine, but avoidable, sources. Low production well sites are abundant and their cumulative $CH_4$ emissions are significant: they account for about one-half (95% CI: 37–75%) of US O&G production site $CH_4$ emissions. The site-level $CH_4$ distribution among these sites is highly skewed, with a small fraction (5%) responsible for a large proportion (~50%) of their total emissions and, on average, $CH_4$ losses occur at high

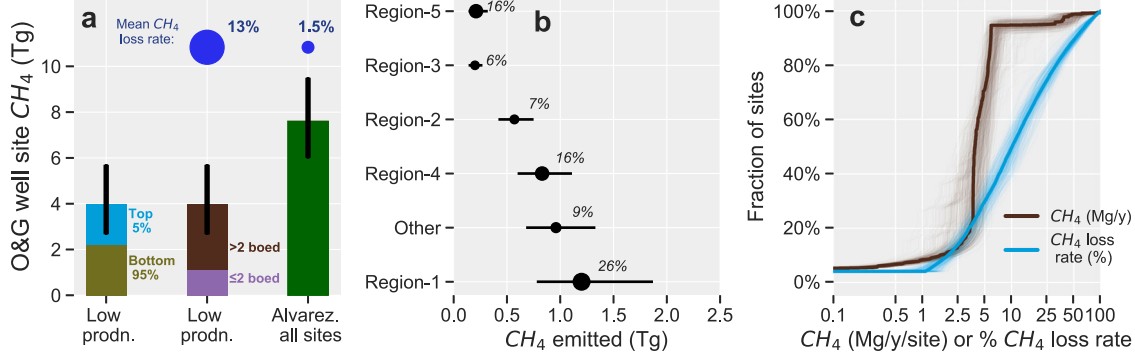

**Fig. 4 National estimate of low production well site $CH_4$ emissions. a** Comparison with Alvarez et al.[16] assessment of total national $CH_4$ emissions from all O&G production sites (Low prodn. = low production sites). Error bars represent the 95% confidence intervals (Methods). The blue bubbles represent the production-normalized $CH_4$ loss rates for low production well sites (this study) and for all O&G sites[16]. **b** Regional estimates of low production well site $CH_4$ emissions (see Fig. 1), with error bars representing the 95% confidence intervals on the mean (Methods). "Other" means total estimates for sites in other locations outside of regions 1–5 in Fig. 1. Symbols are sized by $CH_4$ loss rates relative to gross $CH_4$ production in each region, which are shown as % against each symbol. **c** Modeled distribution of mean site-level $CH_4$ emissions (brown lines) and $CH_4$ loss rates (blue lines). The thick solid lines represent the mean distribution while the thin lines are the results of the 500 simulated distributions for uncertainty assessment (Methods). For visualization, results are shown for the 99% of sites with modeled site-level emissions of up to 100 Mg/year and 100% $CH_4$ loss rates. Additional results in tabular form can be found in Supplementary Tables 5–7. 1 Mg = 1000 kg.

rates exceeding 10% of site-level $CH_4$ production. Identifying high-emitting sites and uncovering the root causes of excessive emissions is key to mitigating $CH_4$ emissions from low production well sites, as is recognizing the disproportionately large role that low producing sites play in contributing to $CH_4$ emissions in the United States.

Field-based observations[10,22] point to avoidable maintenance-related issues as a key driver of $CH_4$ emissions at low production well sites, particularly at older sites that tend to suffer from prolonged lack of attention from their owners or operators. The commonly observed sources of $CH_4$ emissions at these sites, coupled with the stochastic character of high-emission events, suggest routine emissions monitoring and repair has the potential to yield large emission reduction benefits. Ravikumar et al.[34] report that a single LDAR survey reduced site-level emissions by 44% at O&G sites generally, concluding that effective leak mitigation will require frequent surveys utilizing low-cost, rapidly deployable leak detection technologies, such as cheap fixed sensors and fence-line truck-based monitoring. Assuming applicability to low production well sites here, a 44% LDAR effectiveness implies reductions of almost 2 Tg in $CH_4$ emissions after one survey, equivalent to a 24% reduction in total O&G $CH_4$ emissions from all US O&G well sites.

Currently, there is no direct regulation of $CH_4$ emissions from existing low production well sites at the federal level (see Supplementary Table 5 for a summary of state regulatory actions), although the US EPA has recently proposed new regulations that would require quarterly monitoring and repair of $CH_4$ leaks at all well sites that have a potential to emit $CH_4$ emissions >3 metric tons per year as calculated based on bottom-up inventory approaches[35]. Current bottom-up inventory estimates of potential site-level $CH_4$ emissions can underestimate actual emissions, for example, by not adequately accounting for higher emissions due to malfunctions[36]. Our assessment not only underscores the significant contribution of low production well sites to total $CH_4$ from O&G production operations but also supports the inclusion of low production well sites as part of any effective mitigation strategy for O&G $CH_4$ emissions.

As mentioned, routine fugitive emissions monitoring and repair programs inclusive of storage tank fugitives[34,37,38] can be especially effective at these sites, as is mitigating vented emissions, for example, through replacement of high- and low-bleed pneumatic devices with zero-bleed alternatives. The ultralow production cohort of ≤2 boed/site represents a unique challenge given its large size, limited economic value, and proportionally high $CH_4$ emission rates. State and federal policymakers must consider whether and how these well sites can be operated economically while minimizing $CH_4$ emissions, and if they cannot be, how to finance their proper plugging and abandonment.

Current economic support for low production well site owners includes programs from the Internal Revenue Service and several states that incentivize low production well site operations through tax credits that kick in when commodity prices drop below a predetermined threshold[39]. The goals for these programs are to support continued low production well site operation as an alternative to shutting in wells in a low-price environment, but inadvertently incentivize continued emissions of $CH_4$ and other harmful air pollutants linked to O&G operations. Thus, the role of low production well sites needs to be reassessed in light of their outsized importance relative to $CH_4$ emissions from the O&G sector and related mitigation opportunities. As part of this, there is a need for more measurement-based data and a more comprehensive look at the externalities of these low production sites, owned by over 10,000 individuals and small corporations nationally.

## Methods

**Well site O&G data**. We use the monthly O&G well-level and production data available from Enverus Prism[19], aggregating monthly production data for 2019 and deriving average well-level production rates (barrels of oil equivalent per day, boed) based on the reported number of production days (Supplementary Note 1). We use the monthly production data as is, acknowledging there may be uncertainty in the data that are difficult to quantify, for example, due to reporting errors. We filtered the well-level data for active onshore wells ($n = 842,978$) and used geospatial clustering approaches to derive well site attributes (i.e., site-level O&G production rates) from well-level data, assuming wells on the same site are clustered within $r$ buffer radius, where $r = 25$ and 50 m for vertically-drilled and horizontally-drilled wells, respectively (Supplementary Note 1). Based on this approach, we estimate the total number of active onshore low production well sites at 565,000 sites, with an uncertainty of $+2/-5\%$ based on a sensitivity assessment of various choices of buffer radii (Supplementary Note 1). The average number of wells per site is 1.03, 1.9, and 1.2 for low production, non-low production, and all O&G well sites, respectively.

We assess the distribution of site-level O&G production by first classifying the data into four O&G production cohorts based on natural breaks in the data as assessed via the Jenks optimization method. The four cohorts are: (i) >0–2, (ii) 2–5.4, (iii) 5.4–9.7, and (iv) 9.7–15 boed (see Supplementary Note 6 for further discussion).

**Low production well site methane emissions data**. Methane emissions measurements at O&G well sites have typically been performed using either onsite, equipment- or component-level measurement approaches or offsite, downwind measurements. In the former, each potentially $CH_4$ emitting component (e.g., valves, flanges, fittings, etc) is screened and their emissions measured and aggregated to provide an estimate of total site-level emissions. In the latter, $CH_4$ plume concentrations emitted from the O&G well site are taken at an appropriate downwind location using near-real-time concentration measurement instruments; emission rates are then estimated by accounting for the dynamics of plume transport from the source to the measurement point. Some offsite measurement-based studies have used chemical tracers released at known flow rates in close proximity to the known emission source[10] to quantify the $CH_4$ emission rate without the need for plume transport models, which are typically based on Gaussian plume dispersion theory[12,13,20].

Previous studies vary in geography and scope; while some focused on low production well sites, others measured low production well sites as part of a larger measurement campaign that also included non-low production well sites. We assessed each relevant, previously published, peer-reviewed study for $CH_4$ measurement data and selected data for low production well site $CH_4$ emissions based on the following criteria:

(i)   The measurements were focused on quantifying total site-level $CH_4$ emissions,
(ii)  Measurements captured both low and high-emitting sites, and
(iii) Both oil and gas production data were reported for each site where they could be obtained (e.g., based on proprietary data, state-level reports or other reported attributes such as the location of the measured site and date of measurement).

Based on the above criteria, we selected 240 site-level measurement data for low production well sites, with 230 measurements taken from studies by Brantley et al.[20], Omara et al.[10], Robertson et al.[21], Omara et al.[17], Caulton et al.[12], and Robertson et al.[13]. We also include ten new low production well site $CH_4$ measurement data in the Delaware sub-basin of the Permian Basin, based on OTM-33A measurements conducted in January 2020 by the same team that previously reported on site-level $CH_4$ emissions data in this region (Robertson et al.[13]) as part of Environmental Defense Fund's PermianMAP campaign[40]. These datasets are included in Supplementary Data 1. One of the limitations of the ground-based downwind site-level measurement approaches is that the quantification of onsite equipment-level emissions is generally not possible. However, these methods do not require operator-provided access and the site-level data we use herein were obtained without advance operator knowledge.

Each study reported an average measured site-level $CH_4$ emission rate, in addition to O&G production for the month of measurement. Most studies did not report the drilling trajectory for the sampled well sites. However, based on our review of metadata available in a few of the studies[10,12,17,40], we identified 84 vertically-drilled well sites, three horizontally-drilled well sites, and three directionally-drilled well sites. We use the reported data as is, including emissions data that were reported as zeros or below the method detection limits (BDL, 0.036 kg $CH_4$/h for OTM-33A/Gaussian dispersion modeling approaches[20,21] and 0.01 kg $CH_4$/h for tracer flux quantification[10]). For studies that did not report production-normalized $CH_4$ emission rates[12,13,20,21], we compute the $CH_4$ loss rates based on the reported gas production rate and assume an average $CH_4$ content in natural gas of 80% $CH_4$[5]. Additional information on these datasets is provided in Supplementary Note 4.

**Analysis of low production well site methane emissions data**. We begin our assessment by characterizing the representativeness of the measured site-level data

relative to the national population of low production well sites. Given the available data attributes (i.e., site-level emission and production rates), we focus our assessment on (i) geographical diversity, (ii) distribution of site-level production rates, and (iii) distribution of site-level $CH_4$ emissions. Our consolidated sample represents broad spatial coverage as indicated by measurements performed in six major O&G producing regions, including the Appalachian, Uinta, Denver-Julesburg, Upper Green River, Barnett, and the Permian regions (Supplementary Fig. 18). The average gas-to-oil ratio (GOR) for low production sites in these basins ranges from 4 Mcf/barrel to 88 Mcf/barrel, well within the national average of 20 Mcf/barrel. Additionally, all O&G production cohorts (i.e., <2, 2–5.4, 5.4–9.7, and 9.7–15 boed) are represented in the measurement data, where reported site-level production data range from 0.01 to 15 boed. However, the overall production distribution for the measurement sites indicates an oversampling of well sites producing >~5 boed when compared with the distribution for all low production sites nationally (Supplementary Fig. 20). Our emissions modeling approach (described below) accounts for this production distribution as we do not want to bias the modeled $CH_4$ emission rates.

Because the emissions datasets are based on measurements in several basins with unique production and other operational characteristics, we assess whether the emissions distributions from specific basins are statistically similar enough to justify combining the datasets for purposes of estimating national-scale emissions. We assess statistical similarities in site-level $CH_4$ emissions distributions using the Kolmogorov–Smirnov two-sample test, limiting our basin-basin comparison to those basins with $n > 25$ observations, with significance established at $p < 1\%$. This assessment included sites in the Denver-Julesburg ($n = 64$), Upper Green River ($n = 29$), and the Appalachian ($n = 79$) basins. Among these basins, we find statistical similarities and considerable overlap in the empirical site-level $CH_4$ emission distributions (Supplementary Fig. 21 and Supplementary Table 4).

To extrapolate measured site-level $CH_4$ emissions to the total population of sites, we develop a hybrid Monte-Carlo and nonparametric Bayesian regression modeling approach to account for the skewed characteristics of the site-level $CH_4$ data and the influence of the below-detection-limit sites. We begin by reconstructing the empirical distribution of the consolidated dataset via a random bootstrapping procedure, from which we simulate the frequency of finding a below-detection limit (BDL) site and a high-emitting site if the sites were randomly sampled, with replacement, $10^4$ times.

We define high-emitting sites as sites that account for the top 5% of total $CH_4$ emissions. The nonparametric bootstrapping procedure indicates that their percent contribution to total $CH_4$ emissions ($\eta_{high}$) varies from ~20 to 75%, with the 50th percentile of ~50% (Fig. 5a), reflecting uncertainty resulting from a relatively small sample size. For each resampled distribution, we compute the frequency of finding a high-emitting site ($f_{high}$), whose absolute emissions exceed 7.3 kg $CH_4$/h (i.e., the minimum emission rate for the top 5% of sites). We follow a similar procedure to create an emission distribution for the site-level $CH_4$ emission rate for the top 5% of sites, applying resampling weights $1/w_i$ to each high-emission rate, where $w_i$ is the relative contribution of high-emitter $i$ to the total $CH_4$ emissions. In addition, with each nonparametric bootstrap sample, we compute the frequency of finding a site with emissions that are below the detection limit of the measurement methods

(reported as zeros). The frequency distribution for BDL sites ($f_{BDL}$) is shown in Fig. 5c and the distribution for the central estimates of high-emitter emission rates is shown in Fig. 5d.

For the bottom 95% of sites with detected emissions above the detection limit, we apply a nonparametric Bayesian regression model to estimate the mean $CH_4$ emission rates as functions of site-level O&G production. This approach accounts for the potential bias due to oversampling of the higher end of the site-level production distribution (Supplementary Fig. 20) as well as the empirically observed emission trends that are weakly dependent on site-level production (Fig. 3b). We apply a log-transformation to the site-level emissions data and model the distribution assuming a univariate normal likelihood with mean $\mu$ and standard deviation $\sigma$. We model $\mu$ as a linear model with a y-intercept $\alpha$ and a spline basis $\omega$, based on a design matrix incorporating a cubic B-spline with $n = 3$ knots (set at 2 boed—beyond which most high-emitters are observed—and at a minimum and maximum boed of $3 \times 10^{-3}$ and 14.97 boed, respectively). We apply relatively weak priors for $\alpha \sim N(0.1, 0.5)$, $\omega \sim N(-1, 1)$ and $\sigma \sim Exp(1)$. For Bayesian inference, we draw 5000 posterior samples from the posterior distribution using the PyMC3[41] implementation of the No-U-Turn Sampler (NUTS)[42] algorithm, resulting in $\alpha = 0.38$ (94% highest posterior density interval: $-0.25$, 1) and $\sigma = 1.3$ (94% HPD interval: 1.2, 1.5). We use these posterior results to generate predictions of the mean site-level $CH_4$ emissions as functions of O&G production for the bottom 95% of sites, which are shown as a solid-dark red line in Fig. 3b. Additional results and discussion for the nonparametric Bayesian modeling procedure are found in Supplementary Note 5.

We then proceed as follows in extrapolating site-level $CH_4$ emissions to the total population of low production well sites ($m = 565,000$ sites). We randomly sample a frequency ($f_{high}$) of high-emitters from the frequency distribution for the top 5% of high-emitting sites based on absolute $CH_4$ emissions (Fig. 5b). We use $f_{high}$ to compute the total number of sites ($n_1$) that are high-emitting at any one time, restricting our selection to sites with site-level O&G production >2 boed/site beyond which most high-emitters are observed (Fig. 3b). For each high-emitting site, we apply a randomly selected $CH_4$ emission rate from the modeled distribution of high-emitter emissions (central estimates shown in Fig. 5d). The remaining sites ($n_2 = m - n_1$) are the bottom 95% of sites, for which we apply a mean $CH_4$ emission rate to each site based on the binning of the posterior predictions from the Bayesian nonparametric regression into 192 discrete production (boed) cohorts. The predictions for the mean $CH_4$ emission rate for each site in the bottom 95% of sites are randomly drawn 500 times from the results of the posterior distributions. As some sites can have below-detection-limit emissions, we decrement the mean emission rate for each site based on a randomly sampled frequency of BDL sites ($f_{BDL}$). For all $m$ low production well sites, we repeat this procedure 500 times and develop a distribution of total $CH_4$ emissions for (i) the top 5% of sites, (ii) the bottom 95% of sites, and (iii) total $CH_4$ emissions for all sites, accounting for the contribution for the top 5% of sites based on the results of the $10^4$ Lorenz curves generated in Fig. 5a ($\eta_{high}$)(Supplementary Note 5 and Supplementary Figs. 14, 22, 23). Each site's modeled $CH_4$ emissions is multiplied by the total number of reported production days (Supplementary Note 1) to estimate the annual total $CH_4$ emissions.

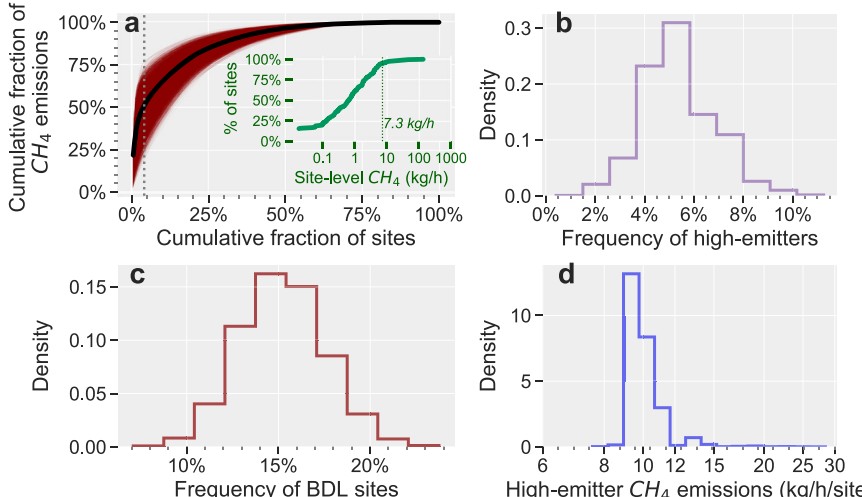

**Fig. 5 Site-level $CH_4$ emission data for low production well sites. a** Lorenz curve showing the cumulative fraction of absolute $CH_4$ emissions as functions of cumulative fraction of sites. The top 5% of sites (dashed vertical line) account for ~50% of total $CH_4$ emissions. The shaded dark red area shows the $10^4$ Lorenz curves derived via a nonparametric bootstrapping of the empirical data, from which the contribution of the top 5% of sites to the total $CH_4$ emissions are obtained ($\eta_{high}$, see Supplementary Fig. 22). Inset is the cumulative distribution function for site-level $CH_4$ emissions, with a dashed vertical line showing the emission rate threshold for the top 5% of high-emitting sites. **b** Histogram of the frequency of finding a high-emitting site based on $10^4$ random bootstrap samples of the empirical data. **c** Histogram of the frequency of below-detection-limit sites. **d** Histogram of the central estimates of high-emitter $CH_4$ leakage rates.

We also assess the same site-level data with a second statistical model that is independent of site-level production rates, following the approach by Zavala-Araiza et al.[7] and assuming the underlying distribution of the site-level $CH_4$ emissions as lognormal. For this assessment, we develop $CH_4$ emissions factors of 3.2 kg $CH_4$/h/site (95% CI: 0.8–18; Supplementary Note 6). The overall results are higher but within 95% confidence intervals of our primary model estimates, which more comprehensively assesses the distribution of emissions relative to the emitter characteristics of the high-emitting sites (top 5% of sites), the bottom 95% of sites with detectable emissions and the below-detection-limit sites.

**Uncertainty assessment**. While available site-level $CH_4$ emissions data are sufficient to derive statistically robust national estimates, we acknowledge the limited sample size ($n = 240$) likely increases uncertainty in our assessment. This uncertainty is driven by variability in measured site-level $CH_4$ emissions, which in turn determines the observed distribution of emissions given the sample size and distribution of site-level production rates. Variability in site-level $CH_4$ emissions distributions might be reasonably expected if more samples were available. Our emissions models for the top 5% of high-emitting sites, the bottom 95% of sites and the BDL sites are based on probabilistic models from which we assess the full range of likely frequency and emissions distributions conditional on the observations (Fig. 5). As described, the mean $CH_4$ emission rate from each of the 565,000 low production site is estimated 500 times in an iterative emissions modeling scheme where both the inputs and outputs are probability distributions reflecting inherent uncertainty in the empirical data. We compute the 95% confidence intervals on our estimates based on the 2.5th and the 97.5th percentiles of the modeled probability distributions for the estimated mean total $CH_4$ emissions. We estimate the mean and 95% confidence intervals on the mean as 2 (1.6–3) and 2 (0.8–3.3) Tg for the bottom 95% and top 5% of sites, respectively. For all low production sites, the combined $CH_4$ distribution has a mean and 95% confidence interval of 4 (3–6) Tg (1 s.f.) as shown in Fig. 4a (see Supplementary Fig. 23 and Supplementary Tables 5–7 for additional details).

## Data availability
All site-level $CH_4$ emission rate data used in this study are included in Supplemental Dataset 1. The national well-level O&G production data comes from Enverus, an O&G software company. Due to its proprietary nature, the data cannot be made openly available. Further information about the data and conditions for access are available at www.enverus.com.

## Code availability
Python 3.7 code used for the data analysis and visualization are available from the authors upon request.

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

## Acknowledgements
This work was made possible by support from the Robertson Foundation. We thank Jevan Yu for assisting with the analysis of the Permian $CH_4$ emissions data from ref. [33]. We are grateful to Adam Peltz and Ramon Alvarez for providing comments. We thank the University of Wyoming team (Shane Murphy's Group), who contributed new OTM-33A measurement data as part of EDF's PermianMAP campaign. We acknowledge the contributions of scores of researchers whose previous works are assessed herein.

## Author contributions
M.O. and S.P.H. conceptualized the study. Formal analysis and visualization were performed by M.O., with contributions to data analysis and interpretation by D.Z.-A., D.R.L., B.H., K.A.R., and S.P.H. M.O. wrote the manuscript with contributions from all co-authors.

## Competing interests
The authors declare no competing interests.
