## [Peer Review File · Nature Communications]

REVIEWER COMMENTS

Reviewer #1 (Remarks to the Author):

The authors report on methane emissions from marginal oil and natural gas wells across the United States. I believe this paper is well-written and the scientific analysis is carefully done. I also believe that the results will be of interest to a broad community and are therefore well-suited for publication in Nature Communications. I particularly appreciated the context and insight that the authors provided about these marginal well sites beyond simply reporting estimated methane emissions from these sites. I recommend publishing this article and have only a few small technical suggestions, detailed below.

Technical suggestions:

Line 32: I think this sentence is missing a comma after "carbon dioxide".

Fig 1: Is it possible to use format the figure as a vector graphic (i.e., postscript, eps, or pdf)? The current figure looks pixelated when I zoom in to look at the details.

Figs 2d, e: What does the x-axis represent in these panels?

Reviewer #2 (Remarks to the Author):

The work produces interesting results--diving into the available data on marginal wells in the U.S. to provide information on the marginal well population.

It relies on available studies and data sets, but analyzes the data to provide interesting information.

The work is transparent.

Reviewer #3 (Remarks to the Author):

What are the noteworthy results?

- Methane emissions from marginal wells are significant and there are a lot of marginal wells. The paper is well-written in general, except a few confusing terminologies as detailed below.

Will the work be of significance to the field and related fields? How does it compare to the established literature? If the work is not original, please provide relevant references.

- Yes, the work is original, although there have been a few papers on marginal wells already.

Does the work support the conclusions and claims, or is additional evidence needed?

- Only 218 measurements are available and analyzed. This number is small relative to the number of marginal wells in the U.S. and we have to question of whether the sample analyzed is representative of U.S.-wide trends.

Are there any flaws in the data analysis, interpretation and conclusions? Do these prohibit publication or require revision?

- There is an overemphasis on the role of the production rate. Other than being the definition of what a marginal well is, I don't think the production rate is a good predictor of a marginal well's methane emissions, which the authors state in the paper as well. Some information on the breakdown of the emissions by the different sources on the site would be helpful, especially when we consider mitigation. And it also begs the question of whether there should be more comparison to abandoned wells.

Is the methodology sound? Does the work meet the expected standards in your field?

- It's unclear how the authors addressed the uncertainty issue and the small sample size (218 out of ~500,000 wells) issues.

Is there enough detail provided in the methods for the work to be reproduced? Yes.

MINOR COMMENTS

line 1: what is the definition of marginal wells? Is the well considered marginal based on production? What do the authors mean by "well" – the well site or the wellhead? I'm not sure if the "marginal oil and gas wells" can be understood by the broad scientific community.

line 18: aren't there marginal wells in the Permian Basin? Which estimate of methane emissions for the Permian Basin is being used here?

line 19: Does "this" refer to the Permian Basin or all marginal wells across the U.S.?

line 64-66: are "low-production" well sites considered marginal in this paper? It's in the next sentence, but I think the introduction to the term, "marginal", is better placed here.

line 85-86: well that produce "very little O&G" are a subset of marginal wells. what is the significance of 2 boed/site?

line 91: is "ultralow-productivity" the same as "very little O&G"?

line 129-131: specify the color used in the panels a to c for these regions.

Fig 2. Panel a: add "operators" after "11,700" and add "sites" after "565,000".

line 150-151: what are the x-axis in the heat maps? What is "Other" - 6 and up? What is "All"? "All" = less than 2 boed/site?

line 155-157: bimodal doesn't mean that the second mode is necessarily higher. It just means that there are two separate distributions - one for sites producing less than 2 boed/site and another for those producing more than 2 boed/site.

line 167: how representative are these 218 measurements of all marginal wells?

line 167: what is considered "non-negligible"?

line 180: Why is the word "absolute" necessary here? Are there non-absolute amounts?

line 182-185 and in general: are the emission sources on marginal sites more similar to active sites or abandoned wells? At the marginal sites, are the well site emissions more important or are other sources (tanks, etc.) more important?

line 195: at marginal wells, are wellheads be the biggest source of emissions?

Fig. 3: Font size in the left panel is larger than in the right panel. Make them consistent. For the y-axis, use nice round numbers so that it's easier to read the plots. Right now, it appears logarithmic but it's unclear and I'm not sure how to interpolate to read the site level CH4 emissions.

line 205: what is the detection limit?

line 220-222: what are the types of emission sources at marginal well sites? are the ones mentioned here relevant to marginal wells?

line 227: I'm not sure if it makes sense to emphasize loss rates for marginal wells. Maybe it

should be treated more like abandoned wells, in which case the literature on abandoned wells should be reviewed and discussed more.

line 265: what is the detection limit?

line 271: here and elsewhere, although I agree that it makes sense to use production define marginal sites, I am not sure that production is the best predictor of marginal site emissions. Overall, there is an over-emphasis in this paper on the role of production rates. This is concerning because one way to interpret this sentence is that lower producing wells are of less concern because they emit less. I don't think this is the message that the authors want to deliver.

line 277: is this 7.6 Tg the current EPA estimate in the GHGI? If so, I would say previously-estimated total O&G CH₄ emissions.

line 295: again, I'm don't think there is a need to emphasize the production-normalized CH₄ loss rates.

Fig. 4, panel b: the % Ch₄ loss rate circles are too similar in size and I cannot tell which number is which. Maybe better to just annotate.

Given the relatively small number of measurements use in the analysis, the confidence intervals seem too narrow. How likely would this interval change with more field measurements?

Fig 4: what is "Marginal*?"

line 311: mention "ultralow" in the first sentence of the paragraph and define it earlier in the paper and be consistent with its usage.

Also, what is the point here, should we be more concerned with ultralow productivity site that low productivity sites?

In the next paragraph you move on the just marginal, which I believe are just low productivity sites. So there needs to be a better transition between this paragraph and the next one. What is the link?

line 340: but do these operators have the funds to do these LDAR surveys? Would many of the marginal sites just get abandoned if such a regulation were to come into play?

line 374: how accurate is the production data for low productivity and ultralow productivity wells?

line 388: ah okay. would have been nice to see this earlier.

line 412: this is a very small data set given that the number of marginal wells are ~500k. I think this point is not emphasized enough in the abstract and throughout the paper. The 218 should definitely be in the abstract. In addition, there needs to be a section on uncertainties. In other words, if another 200+ measurements are available, would you expect the same results as presented here?

line 439: i doubt that the emissions fro the bottom 95% of sites is strongly related to O&G production. And so, I question the applicability of this methodology.

line 486: how was this 30%/-25% determined? For the lower-emitting sites, I can imagine this range to be a lot broader.the revised manuscript. I have no further comments.

REVIEWER COMMENTS

Reviewer #1 (Remarks to the Author):

The authors report on methane emissions from marginal oil and natural gas wells across the United States. I believe this paper is well-written and the scientific analysis is carefully done. I also believe that the results will be of interest to a broad community and are therefore well-suited for publication in Nature Communications. I particularly appreciated the context and insight that the authors provided about these marginal well sites beyond simply reporting estimated methane emissions from these sites. I recommend publishing this article and have only a few small technical suggestions, detailed below.

- We thank Reviewer #1 for this helpful feedback.

Technical suggestions:

Line 32: I think this sentence is missing a comma after "carbon dioxide".

- We have revised this sentence as follows:
“Mitigation of methane (CH₄) emissions, a powerful greenhouse gas with >80× the 20-year warming potential of carbon dioxide, is widely recognized as strategically integral to the attainment of the climate-neutrality goals of the Paris Agreement.”

Fig 1: Is it possible to use format the figure as a vector graphic (i.e., postscript, eps, or pdf)?

The current figure looks pixelated when I zoom in to look at the details.

- Yes, a vector format is available and are provided for all the figures.

Figs 2d, e: What does the x-axis represent in these panels?

- The x-axis represents the regions in Figure 1. We have updated this Figure and included all appropriate labels.

Reviewer #2 (Remarks to the Author):

The work produces interesting results--diving into the available data on marginal wells in the U.S. to provide information on the marginal well population.

It relies on available studies and data sets, but analyzes the data to provide interesting information.

The work is transparent.

- We thank Reviewer #2 for this helpful feedback.

Reviewer #3 (Remarks to the Author):

- We thank Reviewer #3 for these detailed and thorough review of our work, which has helped improve the manuscript. We have carefully considered every point that Reviewer #3 has raised and made specific revisions to the texts and analysis that address these comments. We have provided below our responses to each of the comments, including where we believe no additional revisions are needed.

What are the noteworthy results?

- Methane emissions from marginal wells are significant and there are a lot of marginal wells. The paper is well-written in general, except a few confusing terminologies as detailed below.

Will the work be of significance to the field and related fields? How does it compare to the established literature? If the work is not original, please provide relevant references.

- Yes, the work is original, although there have been a few papers on marginal wells already.

Does the work support the conclusions and claims, or is additional evidence needed?

- Only 218 measurements are available and analyzed. This number is small relative to the number of marginal wells in the U.S. and we have to question of whether the sample analyzed is representative of U.S.-wide trends.

- We agree with Reviewer #3 regarding the limited sample size relative to the total population of low-production well sites in the U.S. However, these datasets are, in our view, the best currently available peer-reviewed datasets on site-level methane emissions from low-production well sites in the U.S. And while sample size is small, we believe they are sufficient to derive statistically robust estimate of national CH₄ emissions from low-production well sites. We show in the revised manuscript that the consolidated data is broadly representative, including (i) measurements are collected from a wide diversity of basins with unique O&G production characteristics, (ii) production distribution are broadly representative of national-scale distribution, with an apparent bias toward sites producing $\sim >5$ boed, and (iii) when we control for sample size, the emission distributions are statistically similar among basins, supporting our consolidation of the emissions and production data to estimate national-scale emissions. We have added texts in the manuscript (Methods and Supplementary Note 8) to make this clearer to the reader.

Are there any flaws in the data analysis, interpretation and conclusions? Do these prohibit publication or require revision?

- There is an overemphasis on the role of the production rate. Other than being the definition of what a marginal well is, I don't think the production rate is a good predictor of a marginal well's methane emissions, which the authors state in the paper as well.

And it also begs the question of whether there should be more comparison to abandoned wells.

- As we indicate in the Main Text, CH₄ emission rates have a weak dependence on site-level production rates; this weak dependence would be expected given the stochastic character of emissions. However, we respectfully disagree that production rate should not be considered in characterizing the emissions from low-production well sites. First, the empirical data suggests emission rates trend lower for sites producing <2 boed (i.e., ultralow-production sites; Fig. 3b in Main Text) but are relatively independent of production rates for sites producing $>\sim 2$ boed (Supplementary Fig. 15). We note that while site-level emission rates appear lower for the ultralow-production rates, the cumulative total emissions from these sites are significant because total emissions are functions of total number of sites (the ultralow-production sites account for $\sim 60\%$ of the total number of low-production sites).

Second, while the empirical distribution of site-level production rates in our sample is broadly representative of the national distribution for low-production sites, there is a slight bias toward sites producing $>\sim 5$ boed (Supplementary Fig. 20). Ignoring production rates in our assessment could lead to a bias in our emission estimates due to an apparent oversampling of this cohort of sites. Our emissions modeling approach—which we show is statistically robust—is informed by these empirical observations,

where we model mean site-level CH_4 emission rates as functions of site-level production rates (for the bottom 95% of sites) while accounting for the stochasticity in emissions that defines higher-emission rates independent of site-level production (i.e., for the top 5% of sites). We do include in our assessment an alternative model that does not depend on production rates and show that the results are within the uncertainty ranges in our assessment. We also note that production rates have been used in previous national-scale emissions assessments by Alvarez et al.¹ and Omara et al.²

Supplementary Fig. 15. Empirical distribution of site-level methane emissions for low-production well sites.

Only sites with reported methane emissions above the method detection limits are shown above.

Supplementary Fig. 20. Distribution of site-level oil and gas production for all low-production well sites (blue, $n = 565,000$), all low-production well sites within the measured basins (orange, $n = 236,000$) shown in Supplementary Fig. 18 and all sampled sites ($n = 218$, green).

Some information on the breakdown of the emissions by the different sources on the site would be helpful, especially when we consider mitigation.

- The studies from which our samples are drawn are site-level measurement studies that did not quantify specific emission sources coming from each measured site. There is, however, abundant evidence that emissions at low-production sites occur from sources that are similar to that at non-low production sites, including intentionally vented emissions and unintentional emissions from well site equipment such as wellheads, pneumatic devices, separators, dehydrators, flare stacks, compressors, and storage vessels. Component- or equipment-level quantification are needed to assess the individual contribution of various emission sources at a given site. Rutherford et al.³ provides a synthesis of component- and equipment-level methane emissions at oil and gas production sites, finding that, in aggregate, tank emissions, equipment leaks, and pneumatic devices were the dominant sources of methane emissions at all well sites accounting for ~90% of total emissions. The authors also estimate that emissions occur at low-production well sites (i.e., <10 Mcfd) from the same sources as at non-low production well sites (> 10 Mcfd), albeit at a comparably lower rate for most sources.

Supplementary Fig. 23. Modeled emission factors for low-production and non-low production sites (Rutherford et al.)

- We have included the following sentences in the Main Text in the section on “Methane emissions at low-production oil and gas well sites: insights from previous site-level studies.”:

“Previous studies indicate CH₄ emissions at low-production well sites arise from sources that are common throughout all O&G production operations, including intentionally vented emissions and unintentional emissions from well site equipment such as wellheads, pneumatic devices, separators, dehydrators, flare stacks, compressors, and storage vessels.”

We also clarify that the site-level quantification approaches used here do not generally resolve source-specific emissions:

“We focus on site-level measurement studies, performed using ground-based downwind measurement approaches that do not require operator-provided access to measured sites and can resolve total CH₄ emissions at each measured site, but generally do not resolve source-specific emissions (Methods).”

And it also begs the question of whether there should be more comparison to abandoned wells.

- We have added the following sentence in the Main Text to clarify what we mean by a low-production well site:

“We consider each low-production site with reported production data as a commercially viable production site or site that routinely produces O&G products that are used for energy consumption. A low-production well site may have one or multiple wellheads

(average 1.03 wells per site; Methods) with O&G processing equipment that may include separators, dehydrators, pneumatic devices, compressors, and/or hydrocarbon liquids storage vessels.”

Abandoned wells are not actively producing facilities and do not have operational process equipment on site that are common to active low-production and non-low production sites (e.g., separators, tanks, pneumatic devices, etc). The wellheads at these sites may be plugged or unplugged. Measured emissions from abandoned wells—which are outside the scope of our study—are substantially lower than that from actively producing well sites. For example, Townsend-Small et al.⁴ estimates an average of 0.01 kg/h for unplugged abandoned wellheads in the U.S. compared with a mean of 0.13 kg/h for ultralow-production wellheads (Deighton et al.⁵) and mean of 0.8 kg/h for all low-production sites in the present study. We have included the following sentences in the revised manuscript:

“The full range of detectable site-level CH₄ emissions at low-production well sites are within that for all O&G production sites but are at least an order of magnitude higher than measured CH₄ emissions at unplugged abandoned wellheads.”

Is the methodology sound? Does the work meet the expected standards in your field?

- It's unclear how the authors addressed the uncertainty issue and the small sample size (218 out of ~500,000 wells) issues.

- We have added a new section in Methods that clarifies the uncertainty estimation approach. The uncertainty is driven by variability in measured site-level CH₄ emissions, which in turn determines the observed distribution of emissions given the sample size and full range of site-level production rates. It is this emissions distribution that would vary if more site-level emissions data were collected. For this reason, our emissions modeling approach is based on probabilistic models from which we assess the full range of likely frequency and emissions distributions conditional on the observations. That is, the frequency of high-emitters, Below-detection-limit (BDL) sites and the emissions for both the bottom 95% of sites and the top 5% of sites are modeled as distributions that reflect inherent uncertainty in the empirical data. As an example, the contribution of the top 5% of sites to cumulative totals in our estimation can vary from ~20% to 75% (Fig. 3a in Main Text).

We have made minor revisions to include random draws of 500 simulated unique emission distributions for the bottom 95% of sites. Previously we estimated each site's methane emissions 250 times based on the mean site-level methane distribution for the bottom 95% of sites (obtained from the Bayesian nonparametric regression model), where variability in total emissions for the bottom 95% of sites was dominated by variability in total number of these sites, which depends on the simulated frequency of BDL and high-emitting sites. In the revised approach, we explicitly simulate 500 unique site-level methane emissions distributions for the bottom 95% of sites, based on random draws from the posterior distribution of the Bayesian model, while still incorporating the emissions and frequency distributions for the top-5% of sites and BDL sites. The mean

estimate is unchanged from previous results, and random draws from these distributions allow us to create emissions distributions for the top 5% of sites and the bottom 95% of sites and assess uncertainties, which are based on the 95% confidence intervals of the distributions, computed as the 2.5th and the 97.5th percentiles.

We estimate the mean and the 95% confidence intervals on the mean as 1.9 (1.5—2.8) Tg and 2.3 (1—3.7) Tg for the bottom 95% and top 5% of sites, respectively. For all low-production sites, the combined CH₄ distribution has a mean of 4.2 (2.8—5.7) Tg. For the bottom 95% of sites, this uncertainty represents a confidence bound of ~-20% to +50% of the mean, and for the top 5%, -57% to +61% of the mean. The combined estimate has a confidence bound of -33% to +36% of the mean, representing a range of 2.8—5.7 Tg of methane emissions. We believe these confidence intervals adequately characterize the uncertainty in our total estimates given the observations and expected variability in empirical emissions distributions.

Supplementary Fig. 22. Distribution of modeled total CH₄ emissions

The 95% confidence intervals are shown in parenthesis and are computed based on the 2.5th and 97.5th percentiles of each distribution of modeled mean total methane emissions.

Is there enough detail provided in the methods for the work to be reproduced? Yes.

MINOR COMMENTS

line 1: what is the definition of marginal wells? Is the well considered marginal based on production? What do the authors mean by “well” – the well site or the wellhead? I’m not sure if the “marginal oil and gas wells” can be understood by the broad scientific community.

- We recognize that the term “marginal” can have multiple meanings, for example, the term may be used to refer to wells that are marginally economic or produces just enough O&G to cover the cost of operations. We have revised the title to avoid any ambiguity: “Methane emissions from U.S. low-production oil and gas well sites.” In addition, we have clarified in the text the difference between a well site and wellhead.

line 18: aren't there marginal wells in the Permian Basin? Which estimate of methane emissions for the Permian Basin is being used here?

- We believe this comparison provides a useful point of reference as the Permian Basin—which does contain low-production sites—is the largest methane emitting basin in the

U.S. We provide additional details for this comparison in the Results and Discussion. The total methane emissions for the Permian Basin that is used for comparison comes from Zhang et al.⁶

line 19: Does "this" refer to the Permian Basin or all marginal wells across the U.S.?

- We have revised this sentence as follows: *“We estimate low-production well sites represent roughly half (37–75%) of all O&G well site CH₄ emissions, and a production-normalized CH₄ loss rate of more than 10%—a factor of 6–12 times higher than the mean CH₄ loss rate of 1.5% for all O&G well sites in the U.S.”*

line 64-66: are "low-production" well sites considered marginal in this paper? It's in the next sentence, but I think the introduction to the term, "marginal", is better placed here.

- We have revised this paragraph to include the definition for “low-production well sites.”

“We define a well site’s total O&G production in units of barrels of oil equivalent per day (boed), a single metric representing the site’s combined oil (barrels produced) and gas (1 boe = 6 thousand cubic feet, Mcf) production averaged over the well site’s total production days in the year. We focus on the low-production well site category, where each site’s combined O&G production rate averaged over the year is ≤15 boed.”

line 85-86: well that produce "very little O&G" are a subset of marginal wells. what is the significance of 2 boed/site?

- We have revised this paragraph to clarify that we classified low-production sites into four cohorts of production rates: (i)>0-2, (ii)2-5.4, (iii)5.4-9.8, and (iv)9.8-15 boed and to define the ultralow-production sites as sites producing <2 boed/site: *“We classify national low-production sites into four cohorts of site-level production rates: (i)>0-2, (ii)2-5.4, (iii)5.4-9.8, and (iv)9.8-15 boed.”*

line 91: is "ultralow-productivity" the same as "very little O&G"?

- Yes, and we have added the following sentence for clarification: *“We refer to these sites as ultralow-production sites and discuss their significance in the following sections.”*

line 129-131: specify the color used in the panels a to c for these regions.

- We have updated Fig. 1 and figure captions per the Reviewer’s suggestions.

Fig 2. Panel a: add "operators" after "11,700" and add "sites" after "565,000".

- We have updated Fig. 2 and figure captions per the Reviewer’s suggestions.

line 150-151: what are the x-axis in the heat maps? What is "Other" - 6 and up? What is "All"? "All" = less than 2 boed/site?

- We have updated the figure caption to clarify the axis labels and other figure captions.

line 155-157: bimodal doesn't mean that the second mode is necessarily higher. It just means that there are two separate distributions - one for sites producing less than 2 boed/site and another for those producing more than 2 boed/site.

- We have revised this sentence as follows: *“For operators with more than 50 operated well sites, a bimodal distribution or a second cluster of sites producing > 2 boed/site emerges.”*

line 167: how representative are these 218 measurements of all marginal wells?

- Please see response above on sample representativeness.

line 167: what is considered "non-negligible"?

- We have removed this description and clarified that site-level *“methane emissions show a wide range of results, reflecting, in part, the stochastic character of CH₄ emissions at these sites.”*

line 180: Why is the word "absolute" necessary here? Are there non-absolute amounts?

- We have revised this sentence as follows: *“The data suggest an increased likelihood for high CH₄ emission potential for low-production well sites producing > ~2 boed/site.”*

Throughout the paper, we make a distinction between absolute CH₄ emissions (mass of methane emitted per hour) and production-normalized or relative CH₄ loss rates (i.e., methane emitted relative to methane produced). We added the following sentences on Page 7 for clarification:

“We assess CH₄ emissions at low-production sites on the basis of absolute CH₄ emission rates (i.e., mass of CH₄ emitted per hour) and the production-normalized CH₄ loss rates (i.e., CH₄ emitted relative to CH₄ production)—a useful metric for comparing the degree of CH₄ loss among different production regions or classes of production sites and can reveal the existence of excessive emissions that may result from avoidable abnormal operating conditions.”

line 182-185 and in general: are the emission sources on marginal sites more similar to active sites or abandoned wells? At the marginal sites, are the well site emissions more important or are other sources (tanks, etc.) more important?

- Please see response above on the comment regarding breakdown of emissions by different sources.

line 195: at marginal wells, are wellheads be the biggest source of emissions?

- Please see response above on the comment regarding breakdown of emissions by different sources.

Fig. 3: Font size in the left panel is larger than in the right panel. Make them consistent.

- We have updated Fig. 3 to make font sizes consistent.

For the y-axis, use nice round numbers so that it's easier to read the plots. Right now, it appears logarithmic but it's unclear and I'm not sure how to interpolate to read the site level CH₄ emissions.

- We have updated the y-axis labels in Fig. 3.

line 205: what is the detection limit?

- We have clarified here and in the Methods that the method detection limits are 0.01 kg/h and 0.036 kg/h for tracer flux and OTM-33A quantification approaches.

line 220-222: what are the types of emission sources at marginal well sites? are the ones mentioned here relevant to marginal wells?

- Please see response above on the comment regarding breakdown of emissions by different sources.

line 227: I'm not sure if it makes sense to emphasize loss rates for marginal wells. Maybe it should be treated more like abandoned wells, in which case the literature on abandoned wells

should be reviewed and discussed more.

- Throughout the manuscript, we present estimates for both absolute emissions (Tg methane emitted) and production-normalized emissions (% of methane produced that is emitted). The loss rates are useful for comparing the degree of CH₄ loss among different production regions or classes of production sites and can reveal the existence of excessive emissions that may result from avoidable abnormal operating conditions.

line 265: what is the detection limit?

- We have included detection limits in the Fig. 3 caption and in the Methods

line 271: here and elsewhere, although I agree that it makes sense to use production define marginal sites, I am not sure that production is the best predictor of marginal site emissions. Overall, there is an over-emphasis in this paper on the role of production rates. This is concerning because one way to interpret this sentence is that lower producing wells are of less concern because they emit less. I don't think this is the message that the authors want to deliver.

- Please see our response above on the role of production rates and how we based our assessment on the empirical observations to obtain statistically robust characterization of the emissions from low-production well sites.

We have also revised this sentence as follows: *“This approach accounts for the empirically observed relative independence of site-level CH₄ emissions with O&G production for sites producing ~>2-4 boed/site and an apparent declining trend in absolute site-level CH₄ emissions for the ultralow-production sites.”*

As we noted before, lower average site-level emissions do not imply total emissions are not concerning. In the Policy Implications section in the manuscript, we especially point out the unique challenge posed by these ultralow-production sites, given their large number, limited economic value and high proportional loss rates.

line 277: is this 7.6 Tg the current EPA estimate in the GHGI? If so, I would say previously estimated total O&G CH₄ emissions.

- The 7.6 Tg cited here comes from Alvarez et al.¹ We have clarified this in the text.

line 295: again, I'm don't think there is a need to emphasize the production-normalized CH₄ loss rates.

- We view this comparison of production-normalized rates as important in contextualizing the degree of methane loss from different categories of production sites in addition to revealing the opportunities for methane mitigation.

Fig. 4, panel b: the % Ch₄ loss rate circles are too similar in size and I cannot tell which number is which. Maybe better to just annotate.

- We have added annotations in Fig. 4 panel b.

Given the relatively small number of measurements use in the analysis, the confidence intervals seem too narrow. How likely would this interval change with more field

measurements?

- Please see our response above on uncertainty assessment.

Fig 4: what is "Marginal*?"

- We have revised Fig. 4a to clarify that the first two bars are estimates for low-production sites from this study.

line 311: mention "ultralow" in the first sentence of the paragraph and define it earlier in the paper and be consistent with its usage.

Also, what is the point here, should we be more concerned with ultralow productivity site that low productivity sites?

- We have defined "ultralow-production" sites early on in the revised manuscript and use this term consistently throughout the manuscript. This entire paragraph underscores the point that ultralow-production sites are especially important in the Appalachian Basin, where they account for over 90% of all low-production sites in the region.

In the next paragraph you move on the just marginal, which I believe are just low productivity sites. So there needs to be a better transition between this paragraph and the next one. What is the link?

- We have moved this section on estimates for operator attribution to the Supplementary Note 9.

line 340: but do these operators have the funds to do these LDAR surveys? Would many of the marginal sites just get abandoned if such a regulation were to come into play?

- Our study is not designed to answer the question of whether operators have the funds to do LDAR surveys. We propose in the section on Policy Implications that *"State and federal policymakers must consider whether and how these well sites can be operated economically while minimizing CH₄ emissions, and if they cannot be, how to finance their proper plugging and abandonment."*

line 374: how accurate is the production data for low productivity and ultralow productivity wells?

- The production data are based on required reporting by operators. We use the data as is and acknowledge there may be uncertainties that are difficult to quantify, for example due to reporting errors.

line 388: ah okay. would have been nice to see this earlier.

- We describe this classification by production cohort in the beginning section on Results and Discussion. *"We classify national low-production sites into four cohorts of site-level production rates: (i)>0-2, (ii)2-5.4, (iii)5.4-9.8, and (iv)9.8-15 boed (See Methods, Supplementary Note 6 for further discussion)."*

line 412: this is a very small data set given that the number of marginal wells are ~500k. I think this point is not emphasized enough in the abstract and throughout the paper. The 218 should definitely be in the abstract. In addition, there needs to be a section on uncertainties. In other words, if another 200+ measurements are available, would you expect the same results as presented here?

- We have revised the Abstract to include the sample size and included the following sentence on Page 7: “While limited in size relative to the total population of low-production sites, these data are drawn from a diversity of O&G production basins and have broadly representative site-level production rates (range: 0.01—15 boed) and distribution that support statistically robust estimation of national-scale CH₄ emissions (Methods).”
- We have added a section on uncertainties, and clarified that our emissions modeling approach is based on probabilistic models that produce distributions of emissions from which we estimate the mean and confidence intervals. Please see the response on Page 5 for a detailed description of the revisions made.

line 439: i doubt that the emissions from the bottom 95% of sites is strongly related to O&G production. And so, I question the applicability of this methodology.

- Please see the response on Page 2 for a detailed discussion on the role of production rates in our models.

line 486: how was this 30%/-25% determined? For the lower-emitting sites, I can imagine this range to be a lot broader.

- Please see the response on Page 5 for a detailed description of the assessment of uncertainties, including that for the bottom-95% of sites and the top 5% of sites.

References

1. Alvarez, R.A. *et al.* Assessment of methane emissions from the U.S. oil and gas supply chain. *Science* **361**, 186-188 (2018).
2. Omara, M. *et al.* Methane emissions from natural gas production sites in the United States: data synthesis and national estimate. *Environ. Sci. Technol.* **52**, 12915-12925 (2018).
3. Rutherford, J.S. *et al.* Closing the methane gap in US oil and natural gas production emissions inventory. *Nat. Comm.* **12**, 4715 (2021).
4. Townsend-Small, A., Ferrara, T., Lyon, D., Fries, A., Lamb, B. Emissions of coalbed and natural gas methane from abandoned oil and gas wells in the United States. *Geophys. Res. Lett.* **43**, 2283–2290 (2016).
5. Deighton, J.A., Townsend-Small, A., Sturmer, S.J., Hoschouer, J., Heldman, L. Measurements show that marginal wells are a disproportionate source of methane relative to production. *J. Air Waste Manag. Assoc.* **70**, 1030-1042 (2020).

REVIEWERS' COMMENTS

Reviewer #3 (Remarks to the Author):

The revisions addresses all comments well and the paper can be published after a few minor edits noted below.

line 109-110: is it fair to make a general statement on the temporal trend of O&G production sites using data for all wells? What fraction of the wells used in this analysis represent horizontally-drilled wells?

line 449: replace "or" with "where they" or something else appropriate.

REVIEWERS' COMMENTS

Reviewer #3 (Remarks to the Author):

The revisions addresses all comments well and the paper can be published after a few minor edits noted below.

We thank the Reviewer for these helpful feedback on the revised version of our manuscript.

line 109-110: is it fair to make a general statement on the temporal trend of O&G production sites using data for all wells? What fraction of the wells used in this analysis represent horizontally-drilled wells?

We have revised this sentence to address the temporal decline in production rates for the horizontally-drilled wells as well as the vertically-drilled wells.

“We find that, on average, the initial site-level production for single-well O&G production sites that are vertically drilled is 20 boed/site, ramping up to ~25 boed/site within the first three months of production, before exponentially declining to below the low-production well site productivity threshold of 15 boed within generally one to two years. For horizontally-drilled wells, we estimate an average initial production of 100 boed/site, ramping up to ~150 boed/site within the first three months and declining to below 15 boed within approximately three to five years (Supplementary Note 3).”

Most measurement studies did not report drilling trajectory for the sampled wells. For studies that reported well identifiers, we reviewed drilling trajectory based on Enverus Prism data and identified 84 vertically drilled well sites, three horizontally-drilled well sites and three directionally-drilled well sites. We have included the following description in the Methods:

“Each study reported an average measured site-level CH₄ emission rate, in addition to O&G production for the month of measurement. Most studies did not report the drilling trajectory for the sampled well sites. However, based on our review of meta data available in a few of the studies,^{10,12,20} we identified 84 vertically-drilled well sites, three horizontally drilled well sites, and three directionally-drilled well sites.”

line 449: replace "or" with "where they" or something else appropriate.

We have revised this sentence as follows:

“Both oil and gas production data were reported for each site where they could be obtained (e.g., based on proprietary data, state-level reports or other reported attributes such as location of the measured site and date of measurement).”